# Is it really a piece of cake to label Geographical Indications with the Nutri-Score? Consumers' behaviour and policy implications

**Alice Stiletto, Samuele Trestini** *

Department TeSAF, University of Padova, Padova, Italy

* samuele.trestini@unipd.it

**Data Availability Statement:** All data files (2 datasets) are available from the "Research DATA UNIPD" database (DOI: 10.25430/researchdata.cab.unipd.it.00000664, URL: http://researchdata.cab.unipd.it/id/eprint/664.).

## Abstract

To improve the dietary habits of the population, the EU, within the Farm to Fork strategy (F2F), is strongly supporting the Nutri-Score (NS) Front Of Pack (FOP) label. Under the NS system, Geographical Indications (GIs) are generally scored as "unhealthy" food, given the predominance of products of animal origin among GIs which are, notoriously, high-fat products. This study aims to determine the impact of the NS label on consumers' preferences for two Protected Designation of Origin (PDO) cheeses, in comparison with generic ones. A Discrete Choice Experiment (DCE) was conducted on 600 Italian consumers through the estimation of a Random Parameter Logit model. Results highlighted that Italian consumers are generally not familiar with the NS and perceive it as a positive characteristic of the product, even if it is signalling an unhealthy choice (D score). However, consumers aware of the Nutri-Score meaning are willing to pay less to buy a product considered "unhealthy" according to this system. Furthermore, we found that consumers who already knew the NS system have homogeneous behaviours in rejecting the product, independently of the association with a PDO certification. This result has important implications on the agri-food sector. If the Nutri-Score becomes mandatory in the EU, consumers might refuse many GIs due to their negative Nutri-Score values. However, the quality of these products is recognized and protected worldwide. In this vein, the GI policy could be questioned by the F2F strategy: both of them aims to reduce information asymmetry producing, at the same time, contrasting results. Within the Geographical Indication policy, the PDO and PGI goods are protected for their quality attributes, which are strictly linked to their geographical origin of the products and traditional know-how. However, the EU adoption of the Nutri-Score could damage these products, reducing their perceived quality/value.

## 1. Introduction

Non-Communicable Diseases (NCDs) are one of the key problems of the XXI century. Among the others, obesity is the main nutritional issue, as it registered an increase of 200% between 1975 and 2016 [1]. As modifiable risk behaviours (e.g., unhealthy diets and physical

**Funding:** The author(s) received no specific funding for this work.

**Competing interests:** The authors have declared that no competing interests exist.

inactivity) are found to be one of the main causes of NCDs, international authorities are currently adopting policy strategies to improve citizens' dietary behaviours to tackle the issue [2].

In this context, informed purchasing choices become a global priority and nutritional labelling has been identified as a crucial aspect in consumer decision making to promote healthier dietary behaviours [3, 4], especially in Mediterranean countries [5]. However, the literature pointed out that consumers pay little attention to nutritional labels. Indeed, although 40% of consumers state they rely on Back of Pack nutritional labels when purchasing [6–8], only 10% actually do so when observed during in-store studies [9]. This might be due to the inconspicuous location of nutritional labels on food packaging [10], consumers' time constraints when shopping [9], and their limited understanding of the Nutrition Facts [11]. It follows that costumers are frequently not able to use this information at the time of purchase. Therefore, a new and easier-to-understand version of nutritional labelling has been widely promoted through Front-Of-Pack (FOP) labels, i.e., graphic labels placed on the front of the package which give concise information about the nutritional profile of the food.

Currently, the EU legislative framework does not regulate FOP nutrition labelling in a harmonized and compulsory way. It follows that multiple FOP schemes co-exist in the European Union [12]. Some of these are promoted by single or groups of countries, such as the Multiple Traffic Lights (MTL) in the United Kingdom and the Green Keyhole in Scandinavian countries. Besides, Reference Intakes (RIs) and summary graded indicators, such as the Nutri-Score, are used by several food manufacturers at EU level. However, as underlined by [13], even if many countries have already adopted different FOP labels, their use on food products packages is still voluntary. The food industry takes advantage of this, since they can endorse the use of FOP labels on many products, but decline to use them when these labels could reduce their sales value [14]. Considering this broad spectrum of FOP labels and their voluntary adoption, the European Commission, within the Farm to Fork strategy, has stressed the need to use a mandatory and self-explanatory Front-Of-Pack nutrition labelling, homogeneous across member states, within 2022.

Among others, the Nutri-Score, tested in a series of experimental and "real-life" studies related to consumers' labels perception [15], understanding [16] and food purchases [17], has proved to be more efficient than other currently available FOP nutritional labels to classify products according to their nutritional quality [15]. In 2018, the European Committee of the Regions called on the European Commission to propose the Nutri-Score as a single mandatory labelling system within the EU. Under the FOP philosophy, the Nutri-Score label has two specific objectives. The first one is to provide consumers with summarized nutritional information in a clear and easy-to-understand way, guiding them towards healthier food choices [18]. The second one appeals to the competition among brands, encouraging the food industry to reformulate their products by improving their nutritional quality, and making them more attractive to consumers [19].

The Nutri-Score FOP label simplifies the identification of the nutritional values of a product conjointly using a chromatic (from green, "healthy", to dark orange, "unhealthy") and an alphabetical scale (from A, "healthy", to E, "unhealthy"). As stressed in the literature, consumers are more able to discriminate against health-related questions about food products in the case of coloured traffic light labels, with respect to other monochromatic ones [20, 21]. Moreover, from a biological perspective, dark orange and green are immediately discerned and discriminated by the human eye [22].

At EU level, this system is currently adopted, on a voluntary basis, in France (October 2017), Belgium (April 2018), Spain (November 2018), Germany (September 2019), the Netherlands (November 2019), and is under consideration in many other countries. It is not used in Italy, and its adoption is an open debate since opinions on it are still controversial. As a consequence, Italian consumers are not familiar with the Nutri-Score because no products with this

FOP label are present in the Italian market yet. Supporting this, it is worth noting that Italian policy makers have proposed the Nutri-Inform battery as an alternative to the Nutri-Score. However, many authors suggested that the FOP labels effectiveness is strictly correlated with consumers' understanding of the label [23]. Hence, as underlined by the European Commission, a mandatory and EU-level implementation of FOP labels would require an assessment of their effectiveness, i.e., their ability to produce the expected results (e.g., reducing the information asymmetry to ensure a more informed—and presumably healthier—choice for the consumer) among all the member states.

Within this framework, a literature analysis highlighted that although several papers (see for instance: [15–17] have been published that outline the power of Nutri-Score to determine the nutritional value of food, very few studies aimed to understand its impact on consumers' purchasing choices [24]. Furthermore, to assess the efficiency of the Nutri-Score at EU level, it is relevant to consider the impact on the demand for products considered of "low health quality". The Nutri-Score classification [25] is based on the average nutritional value of the product, and it considers, per 100 grams of product, the content of nutrients and foods that should be promoted (fibre, protein, fruits and vegetables, for a maximum of 30 points) and the content of nutrients and food that should be limited (energy, saturated fatty acids, sugars, salt, for a maximum of 40 points). Due to the high content of calories and saturated fats, products of animal origin will be characterized by a negative score (i.e., a red or orange label). Contrary to what the MTL does, which focuses only on the nutrients that must be limited in a balanced diet (i.e., fats, saturated fats, sugars and salt), the NS evaluates the overall nutritional quality of the products. Therefore, in the NS case, it is the product itself that is not nutritionally valid, not some of its characteristics.

In this framework, Geographical Indications (GI) products seem to be particularly penalized by the Nutri-Score, given the predominance of products of animal origin in this group. In Italy, which is the top EU country in terms of GIs certifications, 9 out of the top 10 GI products by production value are of animal origin [26] and represent 85% of GI production value and 40% of the national export of products of animal origin. Furthermore, it should be recalled that GI products must follow a strict and traditional product specification and they cannot be easily reformulated to improve their Nutri-Score value. Despite the relevance of this topic, no scientific work has been published to evaluate the impact on consumer preferences of Nutri-Score labels applied on quality products (GI).

In light of this, the present study aims to assess consumers' preferences and their Willingness To Pay (WTP) for Nutri-Score labelled cheeses in Italy (RQ1). We also want to understand if previous knowledge of the label affects the purchasing preferences of the consumers in our sample (RQ2). Then, we want to assess whether the Nutri-Score effect is different on PDO cheeses (RQ3) rather than on generic cheeses. For this purpose, a Discrete Choice Experiment (DCE) was performed on 600 Italian consumers. As consumers familiarity towards different GIs might affect consumers' evaluation of nutritional labels [27], the effect of the GI attribute was estimated using two PDO names with different level of brand familiarity: "Asiago PDO" and "Casatella Trevigiana PDO". The paper is organized as follows: data collection and model specification are provided in the next section. Results are reported in section 3 and discussed in section 4. Some conclusions are drawn in section 5. Finally, in the last section, the policy implications of the present study are reported.

## 2. Data and methods

### 2.1. Experimental design

During the spring 2021, a Discrete Choice Experiment was conducted on 600 Italian consumers, aged over 18, to assess the Nutri-Score effect on their preferences for fresh cheeses in Italy.

The survey, administered online by the Norstat panel agency, was organised according to the scheme in Fig 1. Specifically, we have two different blocks, because of the definition of the PDO (see section 2.2 for more details on this point). In the first block, consumers were presented with an Asiago PDO cheese in comparison with a cheese of the same typology (i.e., fresh cheese of cow milk matured for at least 20 days) without the Designation of Origin. In the second block consumers are presented with Casatella Trevigiana PDO in comparison with a generic casatella cheese. It should be noticed that, in Italy, the term "casatella" is referred to a specific fresh cheese typology, made of whole cow milk with a ripening period of 4–8 days. Respondents of both blocks assure the representativeness of the data according to the following criteria: geographical, gender and age. Descriptive statistics of the sample are summarized in Table 1.

The first part of the questionnaire was focused on understanding consumers' preferences and knowledge of different types of nutritional and FOP labels, as well as customers' purchasing habits for products with these types of labels. 7-points Likert scales were used to address these objectives. The second part of the survey concerns the DCE (see section 2.2). The final section was about consumers' consumption habits for each type of PDO cheese.

## 2.2. Choice experiment

The attributes of the choice experiment and their respective levels are the same for both blocks, except for the price values, as shown in Table 2.

The first attribute taken into account in the experiment is the PDO. Being a Designation of Origin, the PDO sign cannot stand on its own, but it needs to be associated with a registered name. It follows that a specific PDO name needs to be considered in the experiment. However, consumers have different levels of familiarity with different PDO names, or, more precisely, with the different "collective brands" [28]. Literature stressed that brand familiarity may affect consumers' purchasing choices, since if a familiar brand is present on the package, consumers may attach a lower relevance to the FOP nutritional label, thus reducing the efficiency of the label itself [27]. In this sense, PDO names play a role similar to that of private brands. Indeed, Arfini [29] found that consumers perceived the PDO names (i.e., the names of the PDO consortia) as more important than the PDO sign itself. Therefore, PDO names could affect the efficiency of the Nutri-Score label.

To solve this issue, we chose to use two PDO names, with a different degrees of consumers' familiarity: Asiago PDO and Casatella Trevigiana PDO. The first is very renown in Italy, being the fourth Italian PDO cheese for market share. The second, i.e., the Casatella Trevigiana

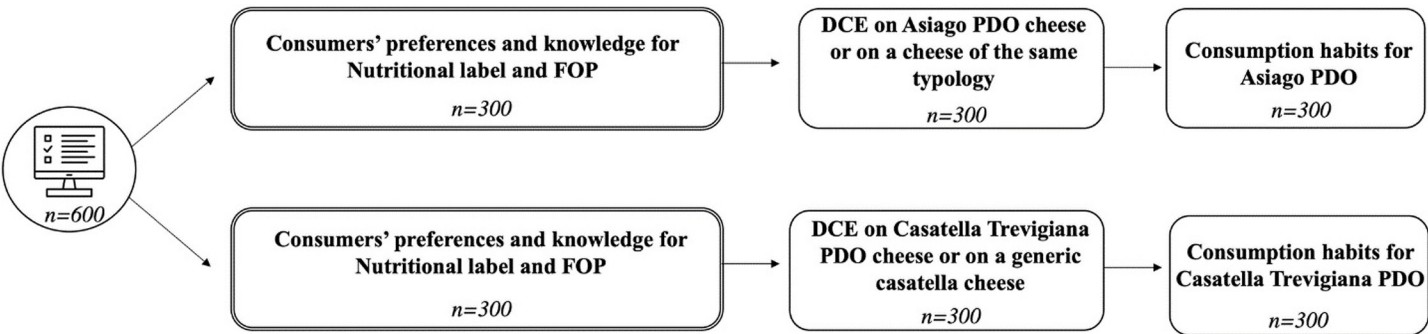

**Fig 1. Structure of the experiment. Note.** The experimental structure is the same in the two blocks. Differences are only found in the product subject of investigation (single line squares). **Source.** Our elaboration.

**Table 1. Descriptive statistics.**

| | | Block 1 | | Block 2 | | |
| | | Sample population | | Sample population | | Italian population |
| Variable | Levels | N. obs | % | N. obs | % | % |
|---|---|---|---|---|---|---|
| Age (years) | 18–24 | 24 | 8.0 | 24 | 8.0 | 8.00 |
| | 25–34 | 39 | 13.0 | 37 | 12.3 | 12.7 |
| | 35–44 | 45 | 15.0 | 46 | 15.3 | 15.3 |
| | 45–54 | 59 | 19.7 | 58 | 19.3 | 19.3 |
| | 55–64 | 49 | 16.3 | 49 | 16.3 | 16.7 |
| | over 65 | 84 | 28.0 | 86 | 28.7 | 28.0 |
| Gender | female | 153 | 51.0 | 153 | 51.0 | 51.3 |
| | male | 147 | 49.0 | 147 | 49.0 | 48.7 |
| Education level | compulsory school | 30 | 10.0 | 27 | 9.0 | 56.0[†] |
| | upper secondary school | 169 | 56.3 | 164 | 54.7 | 26.3[*] |
| | university degree | 69 | 23.0 | 82 | 27.3 | 17.4[*] |
| | post-university degree | 32 | 10.7 | 27 | 9.0 | 0.3[*] |
| | | | | | | **Mean** |
| Family income (€/month) | less than 2,500 | 116 | 38.7 | 121 | 40.3 | 1,627.33[*] €/month |
| | about 2,500 | 137 | 45.7 | 146 | 48.7 | |
| | more than 2,500 | 47 | 15.7 | 33 | 11.0 | |
| Number of household members | | **Mean ± St. Dev. St** | | **Mean ± St. Dev.** | | **Weighted mean** |
| | | 2.84 | 1.15 | 2.83 | 1.28 | 2.35 |

*Note.* [†] In Italy, compulsory schooling is currently not defined by a school cycle, but by reaching the age of 16. Data on compulsory education in Italy are not available on the Eurostat database. Values are estimated.

[*] Eurostat, 2020. (https://ec.europa.eu/eurostat/data/database).

*Source.* ISTAT (Italian National Statistics Institute) and Eurostat.

PDO, is less known and has a sales value 5 times lower than Asiago PDO cheese. Theoretically, we could have considered a PDO attribute with 3 levels: "No PDO sign", "Asiago PDO", and "Casatella PDO". In practice, since the products look different, this was not feasible as we presented real pictures of the alternatives to the respondents (Fig 2), to make the choice experiment closer to a real purchasing situation. It follows that we could not use the same picture to represent the two PDO products and the non-PDO version.

The second attribute considered in the DCE is the Nutri-Score. It was considered as a dummy variable and takes value 1 if this FOP label is present on the product package, 0

**Table 2. Description of the CE attributes and levels.**

| | Block 1 | Block 2 | |
| Attributes | Levels | Levels | Code |
|---|---|---|---|
| PDO name | Asiago PDO | Casatella Trevigiana PDO | (1) |
| | Absence | Absence | (0) |
| Nutri-Score | Score D | Score D | (1) |
| | Absence | Absence | (0) |
| Price | 2.39 €/100g | 2.49 €/100 g | |
| | 2.97 €/100g | 2.91 €/100 g | |
| | 3.55 €/100g | 3.33 €/100 g | |

*Source.* Our elaboration

otherwise. There is no variance within the Nutri-Score level in this DCE, since, according to the Nutri-Score nutrient profiling system (https://www.santepubliquefrance.fr/en/nutri-score), both case studies would be considered as "unhealthy" (score D—orange label). Indeed, Asiago PDO cheese is made of full-cream cow's milk and it has 366 Kcal per 100 g and 24 g of saturated fats. Similarly, the Casatella Trevigiana PDO has 273 Kcal and 23g of saturated fats. It should be noted that the case studies selected in the present analysis are representative of the major GI cheeses in terms of Nutri-Score levels for production values.

Lastly, the price levels were selected based on the current market prices and estimated prices retrieved both from the retail market and online (https://www.miaspesa.it/search) for both products. Price was considered as a continuous variable in the model.

The choice experiment was generated using the R idefix package [30], which used a Modified Federov algorithm to search for efficiency design for discrete choice experiments, based on the multinomial logit model estimates derived from a pilot study conducted on 136 Italian consumers. 12 choice sets were generated, which are thus split into two groups. Six choice sets are therefore assigned to each respondent. Each choice set is represented by two product alternatives (A and B), with different levels of the selected attributes, and a third option (C) that is a no-choice option (Fig 2). The last alternative guarantees a realistic purchasing scenario: in this way, according to Hensher et al. [31], a consumer can choose not to buy the good if its characteristics do not satisfy him/her.

A Random Parameter Model (RPL), which is theoretically explained in the next section, was estimated to assess consumer preferences for Nutri-Score labelled GIs cheeses.

## 2.3. Theoretical explanation of the model and model specification

The DCE method is based on the Random Utility Theory [32], which assumes that a consumer, among the different possibilities provided, chooses the alternative that guarantees him/her the highest utility. According to Lancaster [33], the utility that a consumer derives from buying a product is not related to the product itself, but to the bundle of its attributes, while, according to McFadden [32], the utility is the sum of an observable component and a random error (unobservable) term.

Although the levels of the attributes of each alternative can be observed, the individuals' preferences cannot be directly detected. To assess consumers' preferences expressed through DCE, the McFadden [32] Multinomial Logit (MNL) model is generally adopted, assuming the

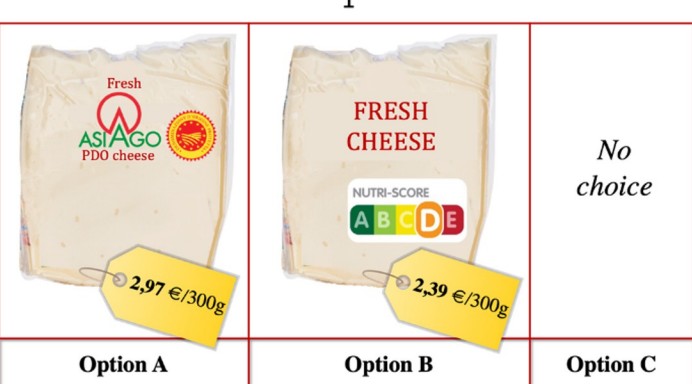 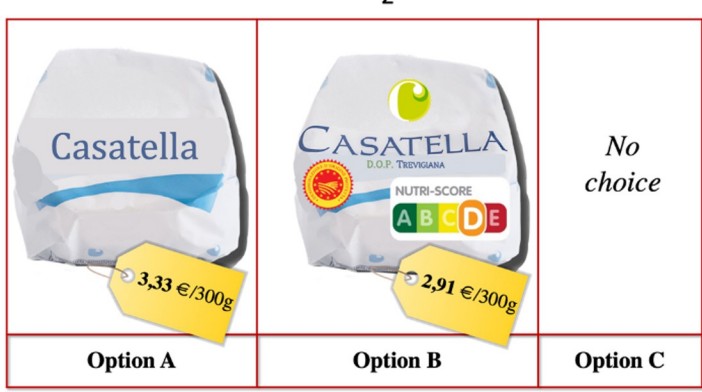

**Fig 2. Example of a choice set for Block 1 (1) and Block 2 (2).** *Note.* The original choice tasks are presented to the respondents in Italian. In the figure, terms are translated in English to facilitated the understanding of the readers. ***Source.*** Our elaboration.

homogeneity in consumers' preferences. However, consumers are assumed to differ in their preferences. Hence, the Random Parameter Logit (RPL) model is implemented in this paper to overcome this restriction [34]. The utility function for the RPL model is described as:

$$U_{nit} = \beta'_n x_{nit} + \varepsilon_{nit} \tag{1}$$

where $\beta_n$ is a vector of coefficients specific of the individual $n$ and $x_{nit}$ is a vector of observed attributes that are related to individual $n$ and alternative $i$ on choice occasion $t$. Given the $\beta_n$ and $x_{nit}$ vectors in (1), the probability that the n$^{th}$ consumer chooses alternative $i$ within a set of $j$ alternatives can be expressed as:

$$P_{ni} = \int \frac{exp\ (\beta' x_{ni})}{\sum_j exp\ (\beta' x_{nj})} f(\beta|\theta)d\beta \tag{2}$$

where $f(\beta|\theta)$ is the density distribution of the $\beta$ coefficient and $\theta$ are the parameters of the distribution [35].

Given this framework, in both case study analyses, three different RPL models were estimated (Fig 3) to address the research questions.

As shown in Fig 3, the first model allows the general effect of the Nutri-Score label on Italian consumers' preferences for the proposed cheeses to be assessed. No additional information about what this FOP label is and how it works were provided to consumers. The Nutri-Score being a novelty in Italy, we assumed that previous knowledge of the Nutri-Score could change consumers' perceptions of the proposed products. For this reason, a second RPL model was estimated. The Nutri-Score variable has been replaced with two different variables ($Nutri_k$ and $Nutri_{dk}$). Specifically, $Nutri_k$ takes value 1 if the Nutri-Score label is present in the choice alternative and the n$^{th}$ consumer knows it, 0 otherwise. On the contrary, $Nutri_{dk}$, takes value 1 if the Nutri-Score label is present in the choice alternative and the n$^{th}$ consumer doesn't know it, 0 otherwise. Finally, to highlight the effect of Nutri-Score on PDO cheeses, a third model was employed. To this end, interactions between Nutri-Score and PDO were considered in the model ($PDO^*Nutri_k$; $Nutri_{dk}$).

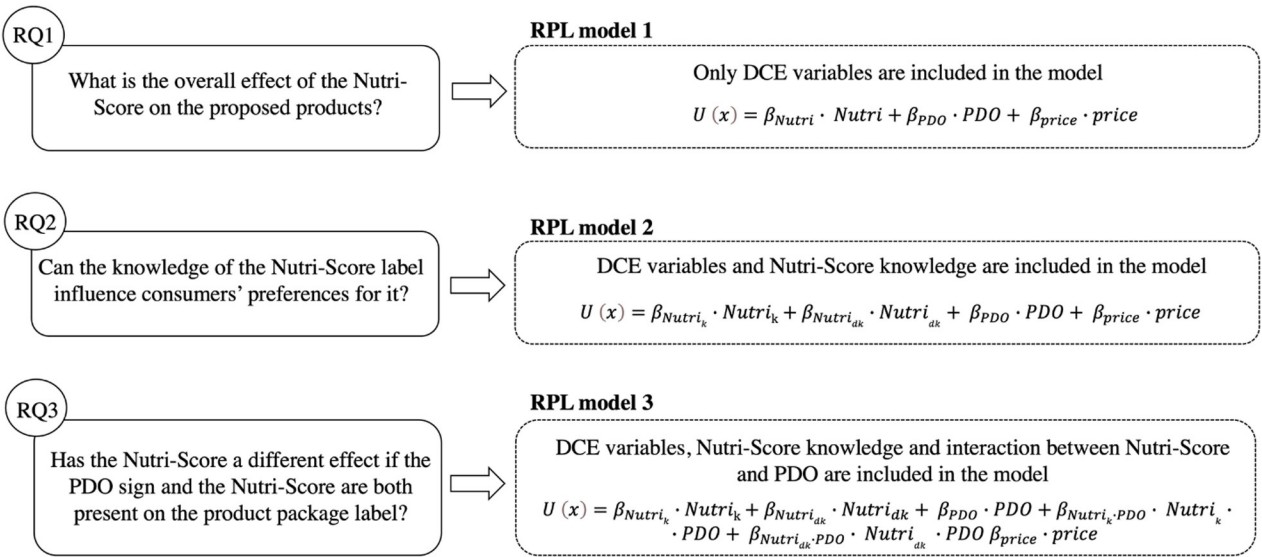

**Fig 3. Model specification.** *Note.* Research questions are reported within the continuous line figures. Specifications of the models that allow to address the RQs are shown, respectively, in the dashed line figures. *Source.* Our elaboration.

## 3. Results

In the following subsection, some descriptive statistics are given to contextualize consumers' consumption habits for the proposed case studies. In addition, consumers' preferences and knowledge of nutritional and FOP labels are reported in Table 3. Results from the estimation of the models are then reported in sections 3.2. and 3.3.

### 3.1. Consumers' habits

In line with their market shares, we found that 89.84% of consumers already knew the Asiago PDO (fresh type) before filling in the DCE; on the contrary, only 22.52% of the consumers in our sample knew the Casatella Trevigiana PDO. Furthermore, we found that consumers' evaluation of the proposed PDO cheeses changed based on their knowledge of the products (Table 3). This is particularly true for the Casatella Trevigiana PDO, for which all the variables investigated in the pair comparison are significantly different between consumers who know the product (*Mean_k*) and those who don't (*Mean_dk*). Specifically, we found that consumers who know the product are more likely to buy it based on its taste, for both cheeses (*Taste*). The same goes for the alleged healthiness of the product (*Healthiness*). However, we interestingly found that consumers who know the Casatella Trevigiana PDO are less prone to buy it than those who don't, since they believe it has a higher calorie (*Calories*) and saturated fat content (*Content of fat*). This is not true for the Asiago PDO case, considering that no differences are found for these variables between the two subgroups of consumers.

Table 4 shows consumer's habits and their knowledge of nutritional and FOP labels. We found that consumers generally declare to consult both the back-package and the front-package nutritional labels, trusting in the European regulation governing products labelling. Among the existing FOP, Multiple Traffic Light (23.29%) and Daily Reference Intake (31.35%) are the best known by the consumers in our sample. On the contrary, only 12.45% of respondents declare to be familiar with the Nutri-Score. Furthermore, we found that, when shopping, consumers mainly rely on nutritional information, price and list of ingredients among the elements reported on the label.

### 3.2. Choice experiment results

**3.2.1. Block 1 (Asiago PDO and generic cheese of the same typology).** Three models were estimated to address the research questions (Fig 2). The model estimates are reported in Table 5 and the relative estimates of consumers' Willingness To Pay in Table 7. Almost all the coefficients are significant at the 5% level. In the three models, the random parameters are assumed to be normally distributed. Price and interaction variables are considered as fixed [36].

**Table 3. Pair comparison of purchase (or not purchase) reasons between consumers who know the product (*Mean_k*) and those who don't (*Mean_dk*).**

| | | Fresh Asiago PDO | | | Casatella Trevigiana PDO | | |
|---|---|---|---|---|---|---|---|
| | | *Mean_k* | *Mean_dk* | *p-value* | *Mean_k* | *Mean_dk* | *p-value* |
| Intention to buy due to: | Taste | 5.81 | 4.26 | 0.000 | 5.90 | 4.63 | 0.000 |
| | Healthiness | 4.91 | 3.81 | 0.000 | 5.26 | 4.26 | 0.000 |
| Intention not to buy due to: | High Calories | 3.84 | 3.81 | 0.657 | 4.38 | 3.39 | 0.000 |
| | High Content of fat | 3.90 | 3.94 | 0.676 | 4.25 | 3.65 | 0.000 |

*Note.* Consumers are asked to express their intention to buy/not to buy the proposed products based on the attributes reported in the table. The scale measure is 1 = totally disagree; 7 = totally agree.

**Table 4. Likert scale measuring consumers' shopping habits for nutritional labelling.**

| Scale items | Mean | St. Dev |
|---|---|---|
| **Attitudes towards nutritional label (Cronbach's Alpha = 0.79)**[*] | | |
| Generally, I don't have much time to read the labels when shopping | 3.31 | 1.89 |
| Too much information on the label confuses me | 3.36 | 1.90 |
| Nutrition labels on the back of the pack are difficult for me to interpret | 3.72 | 1.76 |
| **Consumption habits for nutritional label (Cronbach's Alpha = 0.78)**[*] | | |
| Generally, when shopping, I pay attention to the information on the front of the pack | 5.47 | 1.35 |
| Generally, when shopping, I pay attention to the information on the back of the pack | 5.13 | 1.59 |
| Generally, when shopping, I pay attention to FOP labels, i.e., to the graphical information placed on the front of the pack, which allow consumers to make quick choices. | 4.69 | 1.80 |
| **Attitudes towards information provided on the label (Cronbach's Alpha = 0.90)**[*] | | |
| *When shopping, how important to you are the following elements reported on the label?* | | |
| Nutritional values | 5.48 | 1.36 |
| Price | 5.50 | 1.29 |
| Expiry date or minimum durability date | 5.85 | 1.18 |
| Quantity (grams) | 5.53 | 1.29 |
| Servings per pack | 5.02 | 1.49 |
| Brand | 5.14 | 1.38 |
| Cooking instruction | 4.87 | 1.57 |
| List of ingredients | 5.71 | 1.35 |
| Nutritional claims | 5.53 | 1.36 |
| Health claims | 5.33 | 1.44 |
| Organic certification | 5.16 | 1.65 |
| Environmental sustainability | 5.27 | 1.49 |
| Ethical concern | 4.97 | 1.65 |
| Origin | 5.87 | 1.29 |
| Allergens | 5.13 | 1.72 |
| **Trust (Cronbach's Alpha = 0.72)**[*] | | |
| I trust information provided on the label (Reg. EU 1169/2011) | 5.42 | 1.17 |
| I trust information given in food advertising | 4.44 | 1.48 |
| I trust information given by the EFSA | 5.10 | 1.36 |

*Note.* [*] The scale measure is 1 = totally disagree; 7 = totally agree. Descriptive statistics are performed on the whole sample (n = 600).

*Source.* Our elaboration

**Model 1** (McFadden pseudo $R^2$ = 0.269) allows the overall consumer perception of the Nutri-Score label to be understood. Only the DCE variables are considered in the model. From the results it emerged that both the PDO ($\beta_{PDO}$ = 2.778) and Nutri-Score ($\beta_{Nutri\text{-}Score}$ = 1.492) are, on average, perceived as positive features of the product, despite the Nutri-Score classifying the proposed product as unhealthy. The standard deviation of the Nutri-Score parameter ($\sigma_{Nutri\text{-}Score}$) being statistically different from zero, we can assess, based on the Hole [37] calculation, that 24% of consumers in the sample attached a negative value to the Nutri-Score attribute. To better characterize this latter subsection of consumers, in **Model 2** (McFadden pseudo $R^2$ = 0.270) we measured if previous knowledge of the Nutri-Score has a significant effect on consumers' preferences, thus changing their perception of the Nutri-Score variable. We found that consumers who knew the Nutri-Score label attached a negative value ($\beta$Nutrik = -1.256) to the presence of this FOP on the pack. On the contrary, consumers that didn't

**Table 5. Model estimates for the Fresh Asiago PDO and generic cheese of the same typology case study.**

| | Model 1 | | Model 2 | | Model 3 | |
|---|---|---|---|---|---|---|
| *Random parameter* | | | | | | |
| PDO | 2.778 | *** | 2.673 | *** | 2.684 | *** |
| | (0.535) | | (0.438) | | (0.453) | |
| Nutri-Score | 1.492 | *** | | | | |
| | (0.215) | | | | | |
| Nutri$_k$ | | | -1.256 | *** | -1.253 | *** |
| | | | (0.197) | | (0.212) | |
| Nutri$_{dk}$ | | | 1.577 | *** | 1.605 | *** |
| | | | (0.258) | | (0.257) | |
| *Non-Random parameter* | | | | | | |
| Price | -0.162 | *** | -0.159 | *** | -0.166 | *** |
| | (0.036) | | (0.035) | | (0.037) | |
| PDO* Nutri$_k$ | | | | | -0.099 | n.s. |
| | | | | | (0.308) | |
| PDO* Nutri$_{dk}$ | | | | | -0.240 | n.s. |
| | | | | | (0.268) | |
| *Derived standard deviation of random parameter* | | | | | | |
| PDO | 2.332 | *** | 2.253 | *** | 2.108 | *** |
| | (0.770) | | (0.653) | | (0.700) | |
| Nutri-Score | 2.147 | *** | | | | |
| | (0.613) | | | | | |
| Nutri$_k$ | | | 0.610 | n.s. | 0.841 | n.s. |
| | | | (1.139) | | (0.993) | |
| Nutri$_{dk}$ | | | 2.389 | *** | 2.228 | *** |
| | | | (0.652) | | (0.694) | |
| Number of respondents | 300 | | 300 | | 300 | |
| Number of Obs. | 5400 | | 5400 | | 5400 | |
| Log-likelihood | -1468.7509 | | -1468.1759 | | -1467.7744 | |
| McFadden pseudo R$^2$ | 0.269 | | 0.270 | | 0.270 | |

*Note.* * Significance at 10% level;

** Significance at 5% level;

*** Significance at 1% level. Standard errors in parentheses.

*Source.* Our elaboration.

know the Nutri-Score label were most prone to buy a product with a Nutri-Score label ($\beta$Nutrik = 1.577), despite having a negative score (unhealthy product), compared to a product without the Nutri-Score label.

Furthermore, from the magnitude of the standard deviation ($\sigma_{Nutrik}$ = 0.610), it can be understood that consumers who know the Nutri-Score behave similarly (i.e., the coefficients didn't deviate much from the mean value). There being small variability within individuals' behaviours in this group (*Nutri$_k$*), it can be said that all consumers attribute a negative value to a product labelled with the Nutri-Score (D score—orange label). On the contrary, consumers who didn't knew the Nutri-Score label have a more heterogeneous attitude towards the Nutri-Score.

Finally, the interaction effect of Nutri-Score and PDO on GI cheeses was estimated in **Model 3**. Results showed that the interaction between PDO and Nutri-Score ($\beta_{PDO*Nutrik}$ =

-0.099; $\beta_{PDO*Nutridk}$ = -0.240) were not significantly different from 0. Therefore, the presence of both attributes does not reduce the value perceived by consumers to each attribute.

**3.2.2. Block 2 (Casatella Trevigiana PDO and generic casatella cheese).** As for the block 1, the same models were estimated to answer the RQs about consumers' preferences for the Casatella Trevigiana PDO and a generic casatella. In addition to these, another model (Model 4) was estimated in this case study analysis to better discuss the results as further explained below. The model estimates are reported in Table 6 and the relative estimates of consumers' Willingness To Pay in Table 7. Almost all the coefficients are significant at 5%, except for some interactions in Model 4. The random parameters are assumed to be normally distributed, according to the RPL model assumption. Price and interaction variables are considered as fixed.

**Model 1** (McFadden pseudo $R^2$ = 0.165) summarizes the overall perceptions of consumers about Nutri-Score labelled cheeses. As found for the block 1, consumers attached a positive value to the PDO sign ($\beta_{PDO}$ = 1.735), which is a quality cue of the product. The same goes for the Nutri-Score ($\beta_{Nutri-Score}$ = 0.832), despite having a negative score and signalling an unhealthy product. However, from the magnitude of the standard deviation ($\sigma_{Nutri}$ = *1.56)*, it emerged that 30% of consumers in this sample are not prone to buy a cheese labelled with the Nutri-Score (letter D).

In **Model 2** (McFadden pseudo $R^2$ = 0.164) we try to understand if previous knowledge of the Nutri-Score can influence consumers' perception about this FOP label. As for the block 1, we found that consumers who already knew the Nutri-Score attached a negative value ($\beta_{Nutrik}$ = -1.504) to this label. However, unlike in the case of Asiago PDO and a generic cheese of the same typology, we found that behaviour of the consumers in this sub-group is not homogeneous. Indeed, 30% of consumers who knew the Nutri-Score ($\sigma_{Nutrik}$ = *1.882)* were more prone to buy a product labelled with the Nutri-Score (letter D–orange label) than a product without it. In **Model 3** (McFadden pseudo $R^2$ = 0.171) the effect of the Nutri-Score on PDO cheese was tested. As opposed to what was found for the block 1, it emerged that the interaction between Nutri-Score and PDO is significantly different from 0, both for those consumers who already knew the Nutri-Score and for those who didn't. It means that having both the Nutri-Score and PDO sign on the label alters consumers' perception of these attributes. Specifically, we found that consumers who know the Nutri-Score label attached an additional positive value to a product which has both the Nutri-Score and PDO signs, reducing the negative effect of the Nutri-Score. The opposite effect is estimated when the consumers do not know the Nutri-Score. In **Model 4** (McFadden pseudo $R^2$ = 0.170), it was investigated whether this result is linked to less familiarity of consumers with the Casatella Trevigiana PDO with respect to the Asiago PDO (see Section 3.1). For this purpose, in this model we included interactions among consumers' familiarity with the Casatella PDO (Casatella$_k$), PDO sign, and Nutri-Score label. We found that, for consumers familiar with Casatella Trevigiana PDO and with the Nutri-score ($\beta_{PDO*Nutrik*Casatellak}$ = 0.702), the combined effect of Nutri-Score and PDO was not significantly different from zero, as found for the Asiago PDO. This means that consumers evaluate these attributes (Nutri-Score and PDO) similarly, both when they are labelled separately as well as when they are both on the package. Moreover, from the model estimates, it emerged that consumers not familiar with the Casatella Trevigiana PDO, but who are aware of the Nutri-Score system, reduced their negative attitude towards the Nutri-Score $\beta_{PDO*Nutrik*Casatelladk}$ = 0.872) if the product also carries the PDO logo. On the other hand, we found that for consumers who didn't know the Nutri-Score, the joint effect of the PDO and Nutri-Score was significantly different from the effect of these attributes taken individually. Basically, the Nutri-Score, *per se* ($\beta_{Nutridk}$ = 0.996), had a positive effect, as well as the PDO designation ($\beta_{PDO}$ = 1.761). The synergistic effect of these two attributes is negative,

**Table 6. Model estimates for the Casatella Trevigiana PDO and generic casatella cheese case study.**

| | Model 1 | | Model 2 | | Model 3 | | Model 4 | |
|---|---|---|---|---|---|---|---|---|
| *Random parameter* | | | | | | | | |
| PDO | 1.735 | *** | 1.642 | *** | 1.712 | *** | 1.761 | *** |
| | (0.300) | | (0.250) | | (0.227) | | (0.239) | |
| Nutri-Score | 0.832 | *** | | | | | | |
| | (0.121) | | | | | | | |
| $Nutri_k$ | | | -0.994 | *** | -1.181 | *** | -1.197 | *** |
| | | | (0.260) | | (0.283) | | (0.284) | |
| $Nutri_{dk}$ | | | 0.768 | *** | 0.991 | *** | 0.996 | *** |
| | | | (0.119) | | (0.116) | | (0.119) | |
| *Non-Random parameter* | | | | | | | | |
| Price | -0.095 | *** | -0.092 | *** | -0.150 | *** | -0.146 | *** |
| | (0.031) | | (0.030) | | (0.033) | | (0.033) | |
| $PDO*Nutri_k$ | | | | | 0.719 | ** | | |
| | | | | | (0.315) | | | |
| $PDO*Nutri_{dk}$ | | | | | -0.659 | *** | | |
| | | | | | (0.172) | | | |
| $PDO* Nutri_k *Casatella_k$ | | | | | | | 0.702 | n.s. |
| | | | | | | | (0.453) | |
| $PDO* Nutri_k * Casatella_{dk}$ | | | | | | | 0.872 | ** |
| | | | | | | | (0.422) | |
| $PDO* Nutri_{dk} *Casatella_k$ | | | | | | | -0.698 | ** |
| | | | | | | | (0.283) | |
| $PDO* Nutri_{dk} *Casatella_{dk}$ | | | | | | | -0.628 | *** |
| | | | | | | | (0.185) | |
| *Derived standard deviation of random parameter* | | | | | | | | |
| PDO | 1.591 | ** | 1.421 | ** | 0.908 | n.s. | 1.137 | * |
| | (0.670) | | (0.588) | | (0.658) | | (0.598) | |
| Nutri-Score | 1.566 | *** | | | | | | |
| | (0.457) | | | | | | | |
| $Nutri_k$ | | | 1.882 | ** | 2.175 | *** | 2.123 | *** |
| | | | (0.817) | | (0.717) | | (0.740) | |
| $Nutri_{dk}$ | | | 1.266 | *** | 0.597 | n.s. | 0.711 | n.s. |
| | | | (0.468) | | (0.648) | | (0.607) | |
| Number of respondents | 300 | | 300 | | 300 | | 300 | |
| Number of Obs. | 5400 | | 5400 | | 5400 | | 5400 | |
| Log-likelihood | -1662.6273 | | -1663.3809 | | -1650.6353 | | -1651.8819 | |
| McFadden pseudo $R^2$ | 0.165 | | 0.164 | | 0.171 | | 0.170 | |

*Note.* * Significance at 10% level;

** Significance at 5% level;

*** Significance at 1% level. Standard errors in parentheses.

*Source.* Our elaboration.

regardless of whether consumers know Casatella Trevigiana PDO ($\beta_{PDO*Nutridk* Casatellak}$ = -0.698) or not ($\beta_{PDO*Nutridk* Casatelladk}$ = -0.628).

## 4. Discussion

The results from our experiment were unexpected. Interestingly, we found that, on average, the consumers in our sample showed a positive WTP for the Nutri-Score label, despite its

**Table 7. Marginal Willingness To Pay (WTP) for CE attributes (€/300 g of cheese).**

| | WTP of the consumers in Block 1 | | | WTP of the consumers in Block 2 | | | |
| --- | --- | --- | --- | --- | --- | --- | --- |
| | **Model 1** | **Model 2** | **Model 3** | **Model 1** | **Model 2** | **Model 3** | **Model 4** |
| PDO | 17.15 | 16.78 | 16.18 | 18.25 | 17.86 | 11.44 | 12.03 |
| Nutri-Score | 9.21 | | | 8.75 | | | |
| Nutri_k | | -7.89 | -7.56 | | -10.80 | -7.89 | -8.18 |
| Nutri_dk | | 9.90 | 9.67 | | 8.35 | 6.62 | 6.81 |
| PDO*Nutri_k | | | | | | 4.80 | |
| PDO*Nutri_dk | | | | | | -4.40 | |
| PDO*Nutri_k*Casatella_k | | | | | | | |
| PDO*Nutri_k* Casatella_dk | | | | | | | 5.96 |
| PDO*Nutri_dk*Casatella _k | | | | | | | -4.77 |
| PDO*Nutri_dk *Casatella_dk | | | | | | | -4.29 |

*Note*. Marginal WTP are shown only for significant variables from RPL models.

*Source*. Our elaboration.

signalling an "unhealthy" product (both the case studies in our experiment gain a Nutri-Score value equal to D, "unhealthy"). However, if consumers were discriminated based on their previous knowledge of the Nutri-Score, a polarisation of the preferences emerged. Specifically, when consumers already knew the Nutri-Score system, their attitudes towards the D value of Nutri-Score results in a negative WTP ($Nutri_k$) in both the case studies, as reported in Table 7. Furthermore, from the magnitude of the standard deviation of the parameter ($Nutri_k$), we found that consumers more aware of the Nutri-Score system behaved similarly to each other. On the contrary, consumers who didn't know the Nutri-Score maintained a heterogeneous behaviour towards this label. Our results are in contrast with previous studies [38, 39] and questions the ability of the Nutri-Score to be self-explanatory, as it sets out to be. As pointed out by Santos et al. [23], the reasons behind this result could be sought in the way nutritional information is reported. Indeed, the Nutri-Score does not provide the nutrient-specific composition of the product. It follows that, without additional explanation of how it should be interpreted, the absence of this information can make this FOP label less intuitive at first sight.

Furthermore, as the Nutri-Score is not used yet in Italy, consumers are not familiar with this label. This partially explained why Italian consumers don't have an established behaviour towards the Nutri-Score label. Indeed, only a small group of consumers know the Nutri-Score well enough to make conscious purchasing choices. Our results are consistent with Santos et al. [23] statement: the effectiveness of the FOP label is context-dependent because consumers usually prefer the FOPs previously implemented, at the expense of the new ones, due to familiarity with them. For instance, despite recent studies pointing out that Nutri-Score [17, 39] and Health Star Rating system [40] are the most effective in guiding healthier food choices, Multiple Traffic Light Label (MTL) is found to be the best option to support Portuguese consumers' healthier purchasing choices, due to the greater familiarity with it [23].

In broad terms, it can be assumed that each country prefers a specific FOP label, which consumers in that country are more akin to. However, using different FOP labels across the EU is exactly what the EU try to avoid, considering that the "Farm to Fork" Strategy stressed the need to have a harmonized FOP label among the European Countries within 2022. In our paper, we found, coherently with Fialon et al. [3], that Italian consumers know MTL better (or the monochromatic version, namely the Guideline Daily Amounts) than the other FOP labels. On the contrary, the literature underlined that the Nutri-Score is more appreciated by French

consumers [17]; this is unsurprising, given that France is the "motherland" of this label. In light of this, it is worth noting that the majority of scientific papers focused on the Nutri-Score have been produced in France. Hence, mandatory and EU-level implementation of the Nutri-Score label needs an effective assessment among all the Member States, especially in those in which consumers are less familiar with the label. Alternatively, a more in depth analysis need to be carried out among the Member States to determine which labelling system (or which couple of labels) achieves the EU objectives (i.e., helping consumers to make healthier food choices) in the most effective and efficient way. Medina-Molina and Pérez-González [41], for instance, supports the double use of a summary FOP label (as Nutri-Score) and nutrient-specific ones, as the presence of both of them improves the ability of consumers to choose the healthier options. Once this information gap is filled, the European Commission could evaluate all the proposed labels and choose the best one to adopt.

Furthermore, when more quality attributes are considered, we found that for the less known good, such as the Casatella Trevigiana PDO, prior knowledge of the product is a discriminating factor in consumers' attitude towards Nutri-Score. Indeed, consumers who knew the Nutri-Score label, but were not familiar with the Casatella Trevigiana PDO, positively evaluated a product which had both the PDO designation and Nutri-Score (letter D). This result is at odds with what has been found so far: consumers in this segment are willing to pay 8.18€ less for a product with the Nutri-Score and 12.03€ more for having guaranteed a PDO sign on the pack. However, the presence of both these cues reduced the negative effect of the Nutri-Score by 5.96€ (WTP $_{PDO*Nutri\_k*\ Casatella\_dk}$). This might be due to the information asymmetry on product characteristics. Indeed, we found that consumers who didn't know the Casatella Trevigiana PDO before filling in the survey evaluated this cheese as less caloric and fat than those who knew it previously. When evaluating the healthiness of foods, consumers often make errors, wrongly estimating the calories content, for instance [42]. To overcome this issue, they often rely on product labels to infer food healthiness. However, Schneider and Ghosh [43], found that prior belief in the healthiness of the product (or brand) can alter consumers' trust in FOP labels. It follows that the consumers in this segment, believing that they are dealing with a fresh and low-fat cheese, might not have placed too much trust in the Nutri-Score level that was in contrast with their beliefs. Moreover, in some cases, price and consumption habits might have a greater impact than FOP labels on adjusting consumers' behaviour towards healthier alternatives, according to Boztuğ et al. [44].

## 5. Conclusions

The present work aims to understand consumers' preferences for cheeses labelled with Nutri-Score, depending on the presence of the Designation of Origin in Italy. Despite assessing the efficiency of the labelling systems being of primary importance to achieve the EU objectives [45], to date there are not enough studies supporting the compulsory adoption of the Nutri-Score label. Indeed, to the best of our knowledge, this is the first attempt to assess consumers' attitudes towards GI products labelled with this specific FOP. Interestingly, we found that consumers are on average more prone to buy a product labelled with the Nutri-Score – although the "D" score displayed in the experiment should inform about its negative health features – than a product without this label. Ignoring its meaning, consumers might deem that an additional logo on the label (Nutri-Score) could be a sign of product quality. However, when consumers are aware of its meaning, they change their preferences, drastically reducing their WTP for a product labelled with "D" score in the FOP. Furthermore, we found that consumers belonging to this latter segment display a homogeneous behaviour towards the Nutri-Score, expressing their unanimous rejection of a product considered to be of low nutritional quality,

independently of its association with a PDO certification. This result has important implications on the agri-food sector, especially in the field of GI, and allows consumer behaviour to be "forecast". When the Nutri-Score becomes mandatory among all the EU countries and consumers are more familiar with this labelling, their consumption habits may move in the direction of refusing GI cheeses, in favour of "healthier" substitutes, such as industrial and processed or reformulated foods [19]. This would result in a presumable reduction in the sales value of these products, which are one of the pillars of the quality policy for the food sector in the EU. As reported by Hafner and Pravst [46], the application of the NS on cheeses is a concern in many countries. Indeed, only a few of these products (generic cheeses and not PDO products) can have a positive NS (grade A or B) and, consequently, consumers may consider cheeses as unhealthy foods. However, it is commonly known that high-fat products should be consumed in moderation, thus the lack in the variability of grade of the NS in this category could create misunderstanding among consumers and could negatively affect the sales of the products.

In this context, it is worth to notice that 90% of the top ten Italian GI products for sales value are going to be branded as products to be avoided (score E) or reduced in consumption (score D). This is due to the oversimplification of the Nutri-Score system [47], which restricts the concept of food quality only to the macronutrients of the product, especially if compared with the complexity of the quality concept behind the GI concept. Against this background, the Farm to Fork strategy seems to be at odds with the EU GI policy (Reg. 1151/2021). Within the Geographical Indication policy, the PDO and PGI products are protected for their quality attributes, which are strictly linked to their geographical origin and traditional know-how. However, the EU adoption of the Nutri-Score could damage these products. As found in our paper, the PDO logo doesn't have a halo effect [48] on the general evaluation of the quality of the product, as, on average, consumers don't behave differently when the Nutri-Score is present on the package together with the PDO sign rather than when products are devoid of the Quality Certification. This result strengthens what was stated before: promoting two different and contrasting policies of quality (GI policy and the Farm to Fork strategy) at the same time is like having your cake and eating it.

In conclusion, our results, albeit preliminary, allow us to question the efficiency of the Nutri-Score in guiding consumers' purchasing choices within the general framework of EU food policies. Considering that, in our study, the desired effect (i.e., orienting consumers towards healthier choices) is reached only among those consumers who already knew what the Nutri-Score is and how it works, we can assume that this labelling is not self-explanatory. Therefore, it needs a wider efficiency and efficacy assessment and information effort among the Member States. Further studies are thus needed on this topic to understand whether additional information on the Nutri-Score modifies consumers' purchasing preferences and about the best information strategy to support a coherent EU food policy for quality, health, and rural development. Besides, testing the effect of the NS on GI products compared to other FOPs should be important to address the European Commission' request.

## Limitations

In our study, it was considered only one level of the Nutri-Score (letter D). This could be considered a limitation, since results of the present study cannot be extended for those products bearing a positive Nutri-Score value. However, our choice reflects the real condition of GI cheeses, all of which would receive a negative Nutri-Score value. Furthermore, considering that the sample is representative of the Italian population only in terms of age, gender and

geographical area, the results have limited external validity. Non-Italian consumers, with a different familiarity with the label, may behave differently.

## Acknowledgments

We acknowledge Alison Garside for the language revision of the article and the Norstat agency for their support in the data collection.

## Author Contributions

**Conceptualization:** Samuele Trestini.

**Data curation:** Alice Stiletto.

**Formal analysis:** Alice Stiletto.

**Investigation:** Alice Stiletto.

**Methodology:** Alice Stiletto, Samuele Trestini.

**Supervision:** Samuele Trestini.

**Visualization:** Alice Stiletto.

**Writing – original draft:** Alice Stiletto.

**Writing – review & editing:** Samuele Trestini.

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
