## [Decision Letter · Decision Letter 0]

3 Jun 2022

PONE-D-22-02745Is it really a piece of cake to label Geographical Indications with the Nutri-Score? Consumers’ behaviour and policy implicationsPLOS ONE

Dear Dr. Trestini,

Thank you for submitting your manuscript to PLOS ONE. After careful consideration, we feel that it has merit but does not fully meet PLOS ONE’s publication criteria as it currently stands. Therefore, we invite you to submit a revised version of the manuscript that addresses the points raised during the review process. Please ensure that your decision is justified on PLOS ONE’s publication criteria and not, for example, on novelty or perceived impact.

We look forward to receiving your revised manuscript.

Kind regards,

Zhifeng Gao

Academic Editor

PLOS ONE

**Journal requirements:**

2. You indicated that ethical approval was not necessary for your study. We understand that the framework for ethical oversight requirements for studies of this type may differ depending on the setting and we would appreciate some further clarification regarding your research. Could you please provide further details on why your study is exempt from the need for approval and confirmation from your institutional review board or research ethics committee (e.g., in the form of a letter or email correspondence) that ethics review was not necessary for this study? Please include a copy of the correspondence as an ""Other"" file.

4. Please amend your authorship list in your manuscript file to include author names.

Reviewers' comments:

Reviewer's Responses to Questions

**Comments to the Author**

1. Is the manuscript technically sound, and do the data support the conclusions?

Reviewer #1: Partly

Reviewer #2: Yes

2. Has the statistical analysis been performed appropriately and rigorously? 

Reviewer #1: Yes

Reviewer #2: Yes

3. Have the authors made all data underlying the findings in their manuscript fully available?

Reviewer #1: Yes

Reviewer #2: Yes

4. Is the manuscript presented in an intelligible fashion and written in standard English?

Reviewer #1: Yes

Reviewer #2: Yes

5. Review Comments to the Author

Reviewer #1: Dear Authors, I appreciate your research. In my opinion, it's well written. The data support the conclusions and the statistical analysis has been performed appropriately but I have some some revisions to suggest. The sample is not representative and I think you should better specify the selection criteria and modalities.

In addition, in some parts I think the literature is a bit outdated (e.g. p.9 line 37 the reference is to 2008).

I suggest moving then table 7 to the results part and not the discussions.

I would like you to include some more managerial implications for Italian companies in the light of what is happening abroad.

Reviewer #2: Looking at the interaction between the influences of different types of labels is an interesting topic that should be of interest to many readers. I do though have some comments and questions on the manuscript that are detailed below. I hope they are helpful.

1. I’d probably work on a different title. The ‘cake and eating it’ line in conclusion feels forced.

2. There seem to be some simple writing errors, such as in the first sentence of the abstract, where I assume “UE” is supposed to be “EU.” If not, what is UE?

3. I’m concerned with the idea of equating promoting GI’s with promoting an unhealthy policy. GI’s as you know are not based on healthiness criteria and certainly that isn’t the goal behind the program. So, I wouldn’t push too hard this narrative that they are contrasting policies when they are really policies with very different goals and reasons for being.

4. P. 3. So are you saying the EU’s goal is to get consumers to only buy healthy products? Are you proposing that GI’s are abandoned since they mostly ‘help’ unhealthy foods?

5. As I’m not familiar with the Nutri-Score system I’d like a bit more detail how this single measure is created (I’m more familiar with the MTL). Maybe some more on its strengths/weaknesses relative to the MTL. Maybe some examples of different ratings on GI products?

6. P. 4. L. 111. Be careful with phrasing, you aren’t asking if they are willing to pay for the label, but for the labeled product.

7. I’d like more details on the “external agency” – the group doing this should not be secret.

8. Not clear that Figure 1 adds much beyond the text.

9. P. 5. Why are you using such wide Likert scales?

10. On my copy, there are formatting issues with Table 1.

11. Not sure Table A.1 is needed either.

12. P. 14, L. 338. I’d like more discussion and ideas behind this unexpected result. Certainly looks like the Nutri-Score doesn’t work well without education behind it.

13. P. 16. Again, why are some of these results so different?

14. The conclusions seem broad and overly cast GI’s as being unhealthy (GI is not equal to unhealthy).

15. Towards the end, the paper appears to be more about the Nutri-Score and its abilities and less about its contrast with PDO’s. I wrote a note on my copy asking what the paper was really about.

16. I agree with the limitation of only looking at letter D scored products, especially considering my previous comment about how much the paper seems to be examining the value of the Nutri-Score system. It would be very helpful to know how people respond with some of the other grades – including finding GI’s that aren’t D.

17. You also have more limitations than you’ve noted. Think about these more. Also, what studies should be done next?

6. PLOS authors have the option to publish the peer review history of their article (what does this mean?). If published, this will include your full peer review and any attached files.

Reviewer #1: No

Reviewer #2: No

---

## [Author Response · Author response to Decision Letter 0]

7 Aug 2022

We acknowledge the anonymous reviewers for their useful and precious suggestions that enabled us to greatly improve the quality of our manuscript. Accordingly, we completely revised the original paper just following your suggestions.

In the attached file you can find our point-by-point responses to your comments. Each Reviewer Comment (RC) will be followed by our Answer (A), reported in blue. In the manuscripts, modifies are highlighted in yellow.

---

## [Decision Letter · Decision Letter 1]

19 Oct 2022

Is it really a piece of cake to label Geographical Indications with the Nutri-Score? Consumers’ behaviour and policy implications

PONE-D-22-02745R1

Dear Dr. Trestini,

We’re pleased to inform you that your manuscript has been judged scientifically suitable for publication and will be formally accepted for publication once it meets all outstanding technical requirements.

Kind regards,

Zhifeng Gao

Academic Editor

PLOS ONE

Additional Editor Comments (optional):

**Please make sure to address the minor comments by one of the reviewer in your final draft.**

1. The new sentence in the abstract needs work - "...both of them aims to reduce the information asymmetry producing..." Perhaps remove the word 'the?"

2. On page 14 and 16, need a space between "Table" and "7."

**3. I think you are too hard on yourselves in the Limitations by saying you have "no" external validity. I'd be fine with "limits" external validity.**

Reviewers' comments:

Reviewer's Responses to Questions

**Comments to the Author**

1. If the authors have adequately addressed your comments raised in a previous round of review and you feel that this manuscript is now acceptable for publication, you may indicate that here to bypass the “Comments to the Author” section, enter your conflict of interest statement in the “Confidential to Editor” section, and submit your "Accept" recommendation.

Reviewer #1: All comments have been addressed

Reviewer #2: (No Response)

2. Is the manuscript technically sound, and do the data support the conclusions?

Reviewer #1: Yes

Reviewer #2: Yes

3. Has the statistical analysis been performed appropriately and rigorously? 

Reviewer #1: Yes

Reviewer #2: Yes

4. Have the authors made all data underlying the findings in their manuscript fully available?

Reviewer #1: Yes

Reviewer #2: Yes

5. Is the manuscript presented in an intelligible fashion and written in standard English?

Reviewer #1: Yes

Reviewer #2: Yes

6. Review Comments to the Author

Reviewer #1: Dear Authors,

I appreciate your effors to improve the paper. in my opinion, it can be accepted in the present form.

Reviewer #2: Thank you for taking my comments into consideration. I just have a few, very small comments on the revision.

1. The new sentence in the abstract needs work - "...both of them aims to reduce the information asymmetry producing..." Perhaps remove the rword 'the?"

2. On page 14 and 16, need a space between "Table" and "7."

3. I think you are too hard on yourselves in the Limitations by saying you have "no" external validity. I'd be fine with "limits" external validity.

7. PLOS authors have the option to publish the peer review history of their article (what does this mean?). If published, this will include your full peer review and any attached files.

Reviewer #1: No

Reviewer #2: No

---

## [Editor Report · Acceptance letter]

25 Oct 2022

PONE-D-22-02745R1 

Is it really a piece of cake to label Geographical Indications with the Nutri-Score? Consumers’ behaviour and policy implications 

Dear Dr. Trestini:

I'm pleased to inform you that your manuscript has been deemed suitable for publication in PLOS ONE. Congratulations! Your manuscript is now with our production department. 

Kind regards, 

on behalf of

Dr. Zhifeng Gao 

Academic Editor

PLOS ONE